# DUAL FUSION AUTOENCODER FOR GRAPH CLUSTERING

## ABSTRACT

Clustering of graphs has been an attractive topic in recent years. Recent research has focused on deep fusion graph clustering methods, i.e., fusing two different network structures to enhance the performance of clustering by capturing both graph structure information and node feature information. However, this approach is constrained by the representativeness of the chosen neural network and the choice of the fusion mechanism leads to an unpredictable degree of discretization of the learned graph embeddings. It thus becomes crucial to obtain more compact graph embeddings compatible with the clustering task. In this paper, we propose a new end-to-end fusion, dual fusion autoencoder for graph clustering (DFAC) for deep fusion networks. Our model makes full use of the topology and feature information of the graph and is trained simultaneously by multiple components to obtain better graph embedding. Benefiting from our design of a new dual fusion mechanism, this captures cross-modal good embeddings containing node topology and node feature information. Such a design makes it learn relaxed k-means and performs self-supervised training to improve the quality of graph embeddings while reconstructing the graph structure. By optimizing the training process that is in a unified framework, multiple components are mutually beneficial. Experimental results on six publicly available datasets demonstrate the superiority of the proposed method.

## 1 INTRODUCTION

Many real-world research problems rely on graph data mining skills.Typical applications include community detection (Orgaz & Camacho, 2014), group segmentation (Kim et al., 2022), and functional group discovery in enterprise social networks (Hu et al., 2016). However, the complexity of graph structure poses a serious challenge, where the graph clustering task is one of the most important research topics in the above problem. The purpose of graph clustering is to divide nodes in a graph into disjoint groups.

Some traditional methods (e.g., $k$-means (MacQueen, 1967), DBSCAN (Ester et al., 1996), Graphs (Donath & Hoffman, 1973), GMM (Zivkovic, 2004), etc.) only use the features of the nodes. There are also graph-based models(e.g., spectral clustering (Li et al., 2019; Alshammari & Takatsuka, 2019))that only use graphical information and ignore the characteristics of the nodes. Although some models (Zhang et al., 2019a) use both node features and graph structures, the high dimensionality of the data limits the performance of the model.

With the rise of deep learning, deep clustering methods have gradually come into the limelight. Inspired by the structure of autoencoders, graph autoencoder is proposed for unsupervised representation learning and graph clustering. It can handle more unsupervised tasks compared to GCN, existing methods such as (Kipf & Welling, 2016b), (Hasanzadeh et al., 2019), (Li et al., 2022), (Shi et al., 2020a), (Shi et al., 2020b) and (Ma et al., 2018). However, all of these methods suffer from overfitting problems, resulting in learned embeddings that may not be suitable for the subsequent graph clustering task, thus making graph clustering less effective. Therefore, the top priority is to obtain better goal-oriented graph embeddings.

The emergence of attention mechanisms has brought a turning point to the aforementioned issues. It can automatically capture the weights of adjacent nodes on target nodes through attention mechanisms. Therefore, in this paper, we propose the deep fusion autoencoder for graph clustering (DFAC)

that incorporates the graph attention mechanism. On the one hand, the encoder of this framework consists of a deep fusion of autoencoder (AE) and graph attention network (GAT) networks to alleviate the over-smoothing problem. On the other hand, we newly designed a daul fusion mechanism. Double fusion of graph embeddings from shallow to deep layers leads to better clustering results. The decoder, during the model training process, learns the relaxed $k$-means while reconstructing the topological graph information, and the graph embedding guides the optimization process by using the clustering task as a soft label through a self-supervised training module. Our contributions are as follows:

- We propose a multi-component specific architecture based on fusion networks for graph clustering. Based on this, we propose a new dual fusion mechanism for fusing cross-modal data.

- The learned embeddings of our model have good interpretability, making the model also a qualified representation learning model. At the same time our model is end-to-end, which means that multiple components are performed simultaneously, creating mutual benefits.

- Extensive experimental results on public datasets of different sizes show that we are at the state-of-the-art in performance.

## 2 RELATED WORK

### 2.1 DEEP CLUSTERING ALGORITHMS

Autoencoders have been widely used especially for unsupervised learning tasks. Deep Embedded Clustering is the first unsupervised algorithm to apply autoencoder to clustering. The method uses a stacked noise reduction autoencoder learning method to obtain a hidden representation of high-dimensional data and then optimizes the autoencoder parameters using a defined KL loss. Many subsequent methods have also applied similar structures (Yang et al., 2017; Caron et al., 2018).

Guo et al. (2017) argued that the defined clustering loss corrupts the feature space, resulting in obtaining potential representations that are not feature representative, so they reincorporated the decoder to optimize the reconstruction error and clustering loss. Since then, there have been an increasing number of algorithms based on this deep clustering framework (Wang et al., 2018; Dizaji et al., 2017). However, to the best of our knowledge, these algorithms do not apply to graph data. For graph data, which requires careful mining of graph topology and node feature information, goal-oriented clustering of graph data remains an open problem in the field

### 2.2 GRAPH EMBEDDING

Due to the rise of deep learning, the development of neural networks has also affected graph neural networks. Traditional GNN networks assign weights to neighbor nodes obtained by considering setting or learning (Igarashi et al., 2011; Hernández & Salinas, 2004), but the weighting problem leads to the limitations of numerous algorithms, such as the inability to express the length of the ring graph well resulting in the lack of information about the graph topology.

To solve this problem, the GCN structure emerged (Kipf & Welling, 2017), which mainly applies convolutional operations to the graph structure. It assigns weights by graph topology specific information, which is based on the degree of a node to find the edge weights. The higher the degree of a node, the lower the weight of its edges connected to other nodes. This structure also leads to the fact that GCN cannot assign different weights to each of its neighbors.

In the meantime, researchers have found that encoders composed of a single neural network tend to overlook important information, and thus deep fusion networks have been proposed. Inspired by this work of (Fu et al., 2019), (Bo et al., 2020) and (Tu et al., 2021) are the more representative works of deep fusion networks in the field of graph clustering. Bo et al. (2020) first combines DNN and GCN networks and utilizes delivery operators to achieve fusion at the coding level. Tu et al. (2021), on the other hand, fused AE and GNN networks and designed a unique dynamic fusion mechanism to realize fusion enhancement. However, these methods still impose bad limitations on the clustering performance due to the defects of GCN and GNN itself.

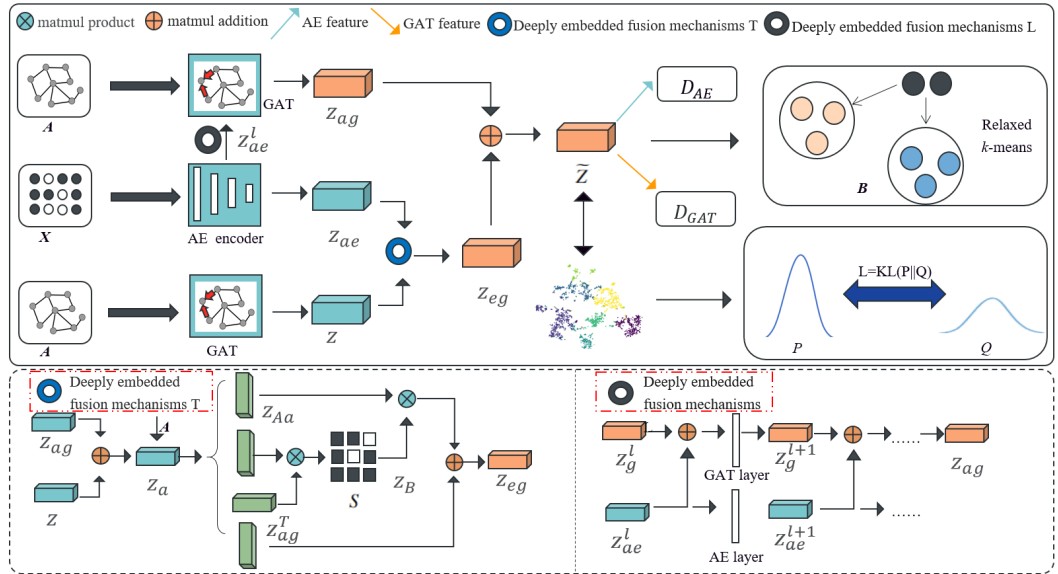

Figure 1: Flowchart of DFAC. Where $A$ represents the structural information of the graph, $X$ is the node feature information in the graph. $B$ process represents the relaxed $k$-means.

The graph attention network (Velickovic et al., 2017) was proposed to solve the above problem of GCN, which can assign different weights to different nodes by Multi-head Attention, and relies on pairs of neighboring nodes during training, rather than on specific network structures. And the scalability is relatively good for directed graphs. It is more reasonable than GCN's update weights which are purely dependent on the graph structure.

## 3 METHOD

In this chapter, we first introduce the organization of our deep fusion network DFAC and then describe how slack $k$-mean clustering and self-supervised clustering lead to and iteratively optimize graph embeddings, respectively. Finally, we summarize the entire framework and define the overall loss(Our framework is shown in Fig 1).

### 3.1 NOTATIONS

This work mainly uses graph data. The graph is represented as $G = (V, E, X, A)$, where the set of nodes is denoted as $V = \{v_1, v_2......v_n\}$, where $v_i$ represents the $i$-th node. The set of edges between nodes is denoted as $E = \{e_{ij}\}$, and the edge between node $i$ and node $j$ is denoted as $e_{ij}$. The topology of the graph G can be represented by the adjacency matrix $A$, where $A_{ij} = 1, e_{ij} \in E$; otherwise, $A_{ij} = 0$. $X = \{x_1, x_2......, x_n\}$ is the node $v_i$ eigenvector attribute value. $x_i \in R^m$.

### 3.2 DEEP FUSION NETWORK

#### 3.2.1 FUSION NETWORK SUBNETWORK

In order to obtain graph embeddings in a unified framework, we developed the deep fusion network as a graph encoder. The idea is to learn the hidden representation of the current node by focusing on the neighbors of each node through GAT. At the same time, the value of attribute $x_i$ containing $v_i$ node's own feature vector will be obtained through AE. Therefore the most straightforward strategy to deal with the neighbors of a node is to integrate the representation of the node with all its neighbors equally. However different neighbor nodes have different importance, which leads to different weights being given to them. Based on the Multi-head Attention mechanism GAT network can be very good to strengthen the important neighbor node weights and weakening the useless neighbor

node weights. The current node $v_i$ hidden representation is calculated as:

$$Z_i^{l+1} = \sigma(\sum_{j \in N_i} \alpha_{ij} W z_i^l) \tag{1}$$

where $\alpha_{ij}$ is the attention factor indicating the importance of neighbor node $v_j$ to node $v_i$ and $\sigma$ is a nonlinear function. $Z_i^{l+1}$ denotes the output representation of node $v_i$ and $N_i$ denotes the neighbors of node $v_i$. Next, we measure the importance of neighbor node $v_j$ in terms of both attribute value and topological distance. The attention coefficient $\alpha_{ij}$ is obtained.

In terms of node attribute values, the $\alpha_{ij}$ can be expressed as a single-layer feedforward neural network for $\vec{x_i}$ and $\vec{x_j}$ in series with a weight vector of $\vec{a} \in R$.

$$d_{ij} = a(W\vec{x_i}, W\vec{x_j}) \tag{2}$$

Note that the coefficients are usually normalized between all neighbors $v_j \in N_i$ with the softmax function, making them easy to compare between nodes.

$$\alpha_{ij} = softmax_j(d_{ij}) = \frac{exp(d_{ij})}{\sum_{k \in N_i} exp(d_{ik})} \tag{3}$$

Next, the current target node needs to be represented by neighboring nodes from the topology. GAT focuses only on the 1-hop neighboring nodes (first-order) of the current node (Velickovic et al., 2017). Since graphs have complex structural relationships, we need higher-order neighboring node relationships. We stack n layers of GAT so that the current node $v_i$ can store information about higher-order neighbors. It can be expressed as:

$$H = \sum_{j \in N_i} \alpha_{ij} v_j = \sum_{j \in N_i} \alpha_{ij}(x_1, x_2......, x_n), x_i \in R^m \tag{4}$$

We choose to reconstruct the graph structure as part of the decoder, which uses the $sigmoid$ function to map $(-\infty, +\infty)$ to the probability space. We minimize the reconstruction error by measuring the difference between $A_i$ and $A_i'$.

$$A_i' = sigmoid(z^t z), L_r = \sum_{i=1}^{n} loss(A_i, A_i') \tag{5}$$

Then there is AE which is the generalized feature extraction method where each layer of AE computes the node features represented as follows,$w^l$ and $b^l$ are hyperparameters. Similar to Eq.(5), we construct the loss function for AE, where $A_a'$ is the graph representation after AE reconstruction.

$$Z_{ae}^l = f(w_l z^{l-1} + b^l), L_{ae} = \sum_{i=1}^{n} ||A_a - A_a'|| \tag{6}$$

Thus the total reconstruction loss is:
$$L_R = L_r + L_{ae} \tag{7}$$

### 3.2.2 DUAL EMBEDDED FUSION MECHANISMS

We apply a new fusion mechanism, which we name dual fusion. The advantage of this mechanism is that it enables the fusion of cross-modal coding from both shallow and deep levels simultaneously so that the learned representation can then adapt to two different kinds of information, i.e., the data itself and the relationship between the data. The overall structure of encoder consists of two GAT networks and an AE encoder network, we call it AGAT. And the dual fusion mechanism part consists of the fusion mechanisms L and T. First we introduce the fusion mechanism L: shallow fusion.

Our GAT network and AE network have the same number of layers, we need to fuse the AE coding of each layer to the GAT coding of the same layer used to increase the hidden features of the nodes. $\epsilon$ is the learnability coefficient, which selectively determines the importance of the two sources

of information based on the nature of the dataset. In this paper, $\epsilon$ was initialized to 0.5 and then automatically adjusted using a gradient fitting method. This gives us the fusion embedding $Z_{ag}^l \in R^{N \times d}$ for each layer. Then we enter it as an input to the next layer of GAT.

$$Z_{ag}^l = (1 - \epsilon)Z^l + \epsilon Z_{ae}^l \tag{8}$$

Next is the deep fusion process T. We are inspired by (Fu et al., 2019) to fuse the features that have been independently encoded by the respective neural networks. First, we need to linearly fuse the features obtained independently by each of them, similarly $\zeta$ is the learnable coefficient, initialized to 0.5, where $Z_a \in R^{N \times d}$:

$$Z_a = (1 - \zeta)Z + \zeta Z_{ae} \tag{9}$$

Then we augment the data with the adjacency matrix of the graph, similar to a message-passing operation. We get $Z_{Aa} = AZ_a$.

Next, we need to calculate the autocorrelation coefficient $S$, where $S_{ij} \in R^{N \times N}$ measures the effect of the $j$-th location on the $i$ location. The more similar the feature representations of two locations are, the greater the correlation between them.

$$S_{ij} = \frac{exp(Z_{Aa}Z_{Aa}^T)_{ij}}{\sum_{k=1}^N exp(Z_{Aa}Z_{Aa}^T)_{ik}} \tag{10}$$

Therefore we perform a correlation operation on $Z_{Aa}$ to obtain the feature $Z_B = SZ_{Aa}, Z_B \in R^{N \times d}$ that takes into account the global sample correlation, we add the augmented data $Z_{Aa}$ and add coefficients to $Z_B$ to balance the performance and to encourage a smooth transfer of information within the fusion mechanism:

$$Z_{eg} = \eta Z_B + Z_{Aa} \tag{11}$$

Note that where $\eta$ is the equilibrium coefficient. We initialize it to 0 and learn its weights when training the network. Finally, we merge the feature $Z_{ag}$ obtained from the shallow fusion process L with the $Z_{eg}$ obtained from the deep fusion process T to complete the final coding fusion of AGAT. $\iota$ is the learnable coefficient, initialized as 0.5.

$$\widetilde{Z} = (1 - \iota)Z_{ag} + \iota Z_{eg} \tag{12}$$

### 3.3 RELAXED $k$-MEANS AND SELF-SUPERVISED EMBEDDING

In this work (Zhang & Li, 2023), Zhang et al. proved that under certain conditions, relaxing $k$-means can obtain the optimal partition with the inner product. They mainly proved that if the following two assumptions hold, and only if $x_i$ and $x_j$ belong to two different clusters, $x_i^T x_j = 0$ is true, then the relaxed $k$-means will give an ideal cluster partition. These two assumptions are: 1) For any two data points $x_i$ and $x_j$, $x_i^T x_j \geq 0$ always holds. 2) Set $\lambda_i^{(m)}$ be the $i$-th largest eigenvalue of $Q^{(m)}$. For any $a$ and $b$, $\lambda_1^{(a)} > \lambda_2^{(b)}$ always holds.

So next we will introduce our variant based on relaxed $k$-means: relaxed $k$-means for the decoder part. Set $g_{ij}$ as an indicator. This indicator is used to determine whether the $i$-th point belongs to the $j$-th class. Specifically, $g_{ij} = 1$ if the $i$-th point is assigned to the $j$-th cluster. otherwise, $g_{ij} = 0$. Clearly, $k$-means exploits an implicit assumption that the Euclidean distance can appropriately describe the scatter of data points. We have $g_{ij} \in \{0, 1\}, \sum_{j=1}^c g_{ij} = 1$.

Thus the problem to be solved by relaxed $k$-means can be expressed as follows, where $\{f_j\}_{j=1}^c$ denotes centroids of $c$ clusters.

$$min_{f_i, g_{ij}} \sum_{i=1}^n \sum_{j=1}^c g_{ij} \|x_i - f_j\|_2^2 \tag{13}$$

Let $g_{ij}$ be the $i, j$-th entry of matrix $G$ and $F = \{f_1, f_2, ..., f_c\} \in R^{dc}$. We need to minimize the following objective function:

$$min_{F,G} \|X_T - FG^T\|_F^2, s.t. g_{vb} \in \{0, 1\}, G1_c = 1_n \tag{14}$$

Inspired by the standardized graph Laplacian matrix in spectral clustering Eq.(13), $G$ can be formulated as follows. Where $\lambda$ is the Lagrangian multiplier, $z$ is the eigenvector, $W$ represents the connection weight and $D$ represents the diagonal matrix which satisfies $(D)_{ii} = \sum_{j=1}^{n} g_{ij} = |\zeta_i|$.

$$D^{-\frac{1}{2}}(D - W)D^{-\frac{1}{2}}z = z\lambda, s.t.z_0 = D^{-\frac{1}{2}}1 \tag{15}$$

$$G = V/D^{-\frac{1}{2}}, s.t.V^t V = I. \tag{16}$$

$I$ denotes the identify matrix. By substituting $V$ for $G$, the objective of $k$-means can be derived as:

$$\begin{aligned} L_c &= ||X^T - FV^T||_F^2 \\ &= tr(X^T X) - 2tr(X^T V F^T) + tr(FV^T V F^T) \end{aligned} \tag{17}$$

Deriving $L_c$, when the equation equals 0 gives:

$$\bigtriangledown_F L_c = -2X^T V + 2FV^T V = 0 \tag{18}$$

After continuing the derivation, we obtain the following equation, and the loss of the relaxed $k$-means part $L_c$ is obtained.

$$F = FV^T V = VX^T \tag{19}$$

$$L_c = tr(X^T X) - tr(V^T X X^T V) \tag{20}$$

In the clustering problem, we consider that the clusters should be made more "separated" from each other and more "tight" within the clusters. Therefore, we use a self-supervision method to optimize the graph embeddings.

For the $i$-th sample and $j$-th category, we use Student's $t$-distribution (van der Maaten & offrey Hinton., 2008) as a measure of the similarity between the data representation $z_i$ and the category centroid vector $\mu_j$.where $v$ is the degree of freedom of Student's $t$-distribution, we take $v = 1$, and $q_{ij}$ is the probability of considering sample $i$ to be assigned to category $j$.

$$q_{ij} = \frac{(1 + ||z_i - \mu_j||^2/v)^{-\frac{v+1}{2}}}{\sum_K (1 + ||z_i - \mu_k||^2/v)^{-\frac{v+1}{2}}} \tag{21}$$

On the other hand, We want to make the data representations closer to the category centers, to improve the cluster cohesion, and therefore calculate the target distribution $p_{ij}$ is the target distribution defined as $p_{ij} = \frac{q_{ij}^2/S_j}{\sum_{j'} q_{ij'}^2/S_{j'}}$. $S_j$ is the summation of $q_{ij}$ in the $j$-th category. Thus $S_j = \sum_i q_{ij}$.The soft assignment of the fusion embedding of the subnetworks of AGAT is computed separately using the formula above. We label the one obtained by the L fusion process as $Q^l$ and the one obtained by T as $Q^t$. At this point, we label all Q distributions uniformly as $\widetilde{Q}$. In the target distribution P, each value of $\widetilde{Q}$ is squared and then normalized so that it ultimately has a higher confidence level. To minimize the clustering loss, we construct the loss using the KL scatter:

$$L_s = KL(P||\widetilde{Q}) = \sum_i \sum_j p_{ij} log \frac{p_{ij}}{(q_{ij} + q_{ij}^l + q_{ij}^t)/3} \tag{22}$$

Eq. (22) represents the method's simultaneous alignment of the dual fusion mechanism with the soft assignment distribution of the fusion representations generated by its sub-networks L and T processes with the robust target distribution. This forms a well-established self-supervised strategy that effectively optimizes the fusion embedding. We will explore this part in depth in subsequent ablation experiments.

## 3.4 JOINT EMBEDDING AND CLUSTERING

We co-optimize graph embedding and clustering learning by combining these multiple components to learn together, defining the total objective function as

$$L = L_R + \alpha L_c + \beta L_s \tag{23}$$

where $\alpha, \beta > 0$ are the coefficients that control the balance between the two. It should be emphasized that, since a central idea of unsupervised neural networks is to define the loss of training through reconstruction, $L_c$ and $L_s$ provide a new way to train neural networks unsupervised.

---

**Algorithm 1** Deep attention embedded graph autoencoder (DFAC)

---

**Input:** Graph $G$ with $n$ nodes; Number of iterations $Iter$; Number of clusters $k$;Number of layers $La$ and dimensions of fusion network $x_n$.

**Output:** Clustering results and hidden embedding $\widetilde{Z}$.

    The autoencoder is updated by minimizing $L_R$ to obtain the initial hidden embedding $\widetilde{Z}$ of the autoencoder and calculate the initial clustering center $\mu$.

    **while** max iterations$< iter$ or convergence **do**

        **repeat:**Update the $P, \widetilde{Q}, \widetilde{Z}, \mu$ and calculate $L$.

            Update $W$ by the gradient descent.

        **until:**Converge or exceed the maximum iteration.

        Cluster and clustering evaluation metrics obtained by relaxed $k$-means.

---

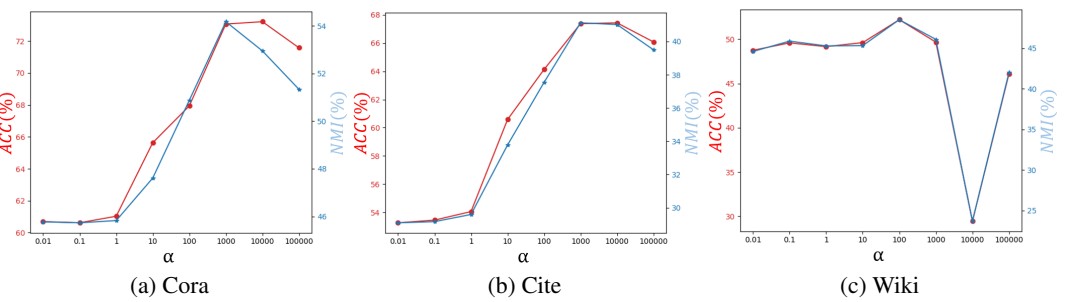

| (a) Cora | (b) Cite | (c) Wiki |

Figure 2: The variation of ACC and NMI on each data set for different values of $\alpha$

## 4 EXPERIMENTS

### 4.1 PARAMETER SETTINGS

In our experiments, we chose a two-layer GAT encoder structure. Both activation functions of the two layers are ReLU. $\alpha$ and $\beta$ are two important parameters that can have a clear impact on the experimental results. $\alpha$ is searched in the range $\{10^{-2}, 10^{-1}, 10^0, 10^1, 10^2, 10^3, 10^4, 10^5\}$. The final choice is shown below Fig 2. $\beta$ is chosen in the same way, here we choose 10. The maximum number of training iterations is set to 200. After pre-training, the fine-tuning learning rate is set to $10^{-4}$. The maximum number of iterations for updating $P$ is set to 30, and the maximum number of internal iterations for updating the neural network is set to 5. The dimensionality of the embedding layer is set to 128, the reason for this is that the embedding dimension may lead to slow convergence of DFAC and difficult training if it is equal to the number of clusters. However, it has to be said that the problem of choosing hyperparameters in similar experiments is still to be solved. All codes are implemented by torch-1.13.1 on a server with four NVIDIA GeForce RTX 3080 GPUs.

### 4.2 COMPARISON WITH STATE-OF-THE-ART METHODS

In our experiments, we evaluated the proposed algorithm on 6 popular public datasets, including 3 graph datasets (Cora (Sen et al., 2008), Cite (Sen et al., 2008), and Wiki (Yang et al., 2015b)) and 2 non-graph datasets (USPS (Hull, 1994), HHAR (Li et al., 2021)), and one dataset with larger size (ogbn-arxiv (Hu et al., 2020)), see Table 1.For datasets with missing adjacency matrices, we followed (Bo et al., 2020) and constructed the matrices using the heat kernel method.

In our experiments, we compared 14 algorithms with our approach:$k$-means (MacQueen, 1967), Graph-encoder (Tian et al., 2014), Deep Walk (Perozzi et al., 2014), DNGR (Cao et al., 2016), TADW (Yang et al., 2015a), GAE (Kipf & Welling, 2016a), ARGE (Pan et al., 2018), ARVGE (Pan et al., 2018), AGC (Zhang et al., 2019b), DAEGC (Wang et al., 2019), EGAE (Zhang & Li, 2023), SDCN (Bo et al., 2020), DFCN (Tu et al., 2021), S3GC (Devvrit et al., 2022). In this paper, three metrics, clustering accuracy (ACC), normalized mutual information (NMI) and adjusted rand index (ARI), are used to verify the performance of various models.

Table 1: Information of dataset

| Dataset | Nodes | Features | Clusters | Links |
|---|---|---|---|---|
| Cora | 2708 | 1433 | 7 | 5429 |
| Cite | 3312 | 3703 | 6 | 4732 |
| Wiki | 2405 | 4973 | 17 | 17981 |
| USPS | 9298 | 256 | 10 | * |
| HHAR | 10299 | 561 | 6 | * |
| Ogbn-arxiv | 169343 | 128 | 40 | 1166243 |

Table 2: clustering results(%)

| Methods | Cora | | | Cite | | | Wiki | | | USPS | | | HHAR | | | Ogbn-arxiv | | |
|---|---|---|---|---|---|---|---|---|---|---|---|---|---|---|---|---|---|---|
| | ACC | NMI | ARI | ACC | NMI | ARI | ACC | NMI | ARI | ACC | NMI | ARI | ACC | NMI | ARI | ACC | NMI | ARI |
| $k$-means | 49.62 | 31.78 | 22.92 | 54.00 | 30.80 | 26.96 | 38.82 | 43.01 | 14.89 | 66.80 | 62.60 | 54.61 | 60.00 | 58.90 | 45.10 | 17.60 | 21.60 | 7.40 |
| Graph-Encoder | 32.29 | 8.99 | 1.25 | 21.97 | 2.81 | 1.00 | 18.34 | 18.18 | 0.98 | 35.10 | 10.00 | 11.39 | 38.95 | 12.83 | 9.77 | 8.99 | 13.03 | 3.01 |
| DeepWalk | 50.03 | 36.70 | 28.81 | 36.18 | 3.01 | 10.22 | 37.34 | 30.14 | 17.05 | 47.56 | 27.11 | 18.80 | 46.18 | 34.63 | 25.58 | 16.84 | 24.11 | 11.08 |
| DNGR | 41.90 | 31.84 | 14.21 | 32.59 | 18.01 | 4.30 | 37.58 | 35.84 | 17.95 | 43.15 | 32.30 | 21.00 | 39.17 | 33.01 | 18.90 | * | * | * |
| TADW | 54.55 | 38.20 | 28.00 | 51.52 | 30.87 | 28.47 | 31.01 | 25.07 | 4.23 | 51.20 | 42.00 | 24.19 | 50.31 | 37.01 | 22.20 | * | * | * |
| ARGE | 63.91 | 43.19 | 35.00 | 57.31 | 34.96 | 33.01 | 38.01 | 34.17 | 10.09 | 62.09 | 45.40 | 30.80 | 59.02 | 44.03 | 30.00 | * | * | * |
| ARVGE | 62.33 | 44.00 | 36.01 | 50.30 | 22.12 | 23.06 | 38.34 | 32.81 | 10.63 | 61.79 | 44.10 | 27.26 | 60.00 | 39.83 | 34.05 | * | * | * |
| GAE | 58.18 | 40.39 | 32.88 | 37.91 | 17.04 | 18.25 | 32.83 | 28.59 | 6.18 | 63.11 | 60.0 | 49.92 | 62.00 | 54.03 | 41.50 | 21.01 | 22.03 | 10.91 |
| AGC | 68.92 | 53.86 | —— | 67.00 | 41.13 | —— | 47.65 | 45.28 | —— | 67.25 | 49.16 | —— | 66.99 | 50.00 | —— | * | * | * |
| DAEGC | 68.95 | 34.78 | 36.82 | 66.29 | 38.91 | 40.03 | 49.10 | 28.05 | 32.80 | 73.62 | 71.10 | 63.30 | 75.00 | 68.50 | 59.90 | 30.40 | 40.21 | 23.51 |
| EGAE | 71.53 | 48.47 | 50.26 | 66.58 | 40.11 | 40.99 | 51.19 | 47.47 | 33.07 | 74.97 | 70.70 | 65.26 | 78.53 | 69.22 | 60.63 | 33.10 | 39.20 | 24.90 |
| S3GC | 73.22 | 58.87 | **54.42** | 67.81 | 44.11 | 49.28 | 48.99 | 43.20 | 35.16 | 72.04 | 64.10 | 59.89 | 77.00 | 65.03 | 59.00 | 35.00 | **46.30** | 27.00 |
| SDCN | 71.48 | 46.30 | 40.33 | 66.30 | 39.91 | 40.10 | 51.21 | 40.13 | 30.91 | 78.11 | 79.62 | 71.70 | 84.00 | 80.00 | 72.70 | 34.02 | 40.00 | 22.14 |
| DFCN | 72.16 | 50.89 | 49.10 | 69.51 | 43.52 | 45.53 | 51.08 | 39.85 | 33.07 | 79.40 | 82.70 | **75.45** | 86.01 | 81.38 | 76.13 | 34.70 | 41.97 | 26.09 |
| DFAC | **74.18** | **58.91** | 53.24 | **69.75** | **44.40** | **45.58** | **53.05** | **48.69** | **35.15** | 79.91 | **83.29** | 74.10 | **88.01** | **82.05** | **77.93** | **35.08** | 45.00 | **27.12** |

The results of the experiments on the three datasets are shown in Table 2. We can see that our method clearly outperforms all baselines in most of the evaluation metrics. This proves that just using node features and topology information of the graph leads to poor performance. In contrast, DFAC makes full use of both information to achieve consensus representation learning, which greatly improves the clustering performance. DFAC not only realizes the dynamic interaction between the graph structure and node attributes but also employs a new self-supervised strategy that provides precise guidance for network training. For example, on the Wiki dataset, ACC improves by about 2% and NMI improves by about 2%. It should be noted that * in the table represents the algorithm calculation process exceeding memory.

## 4.3 ABLATION STUDY

We first explored the impact of each module on the overall model performance. We replace the GAT with the GCN autoencoder, denoted as DFAC-GCN; remove the self-supervision module, denoted as DFAC-selfOE; and replace the relaxed $k$-means with the GMM clustering, DFAC-GMM, respectively. From the results Table 3, we can see that the self-supervision module has a limited impact on the algorithm as a whole. Among them, DFAC-GMM has the worst result. Therefore the relaxed $k$-means part plays the biggest role in the overall algorithm.

We then included further exploration of the deep fusion network to investigate the practical usefulness of our proposed new fusion mechanism. We refer to the experimental model using only the shallow fusion mechanism as DFAC-L, and the experimental model using only the deep fusion mechanism as DFAC-T. From the results Fig 3(a)-(f), it seems that our new mechanism that unifies the deep and shallow fusion mechanisms works better, and the DFAC-T that applies the deep fusion mechanism is more effective in comparison to the two.

Finally, as in Fig 3(g)(h), we delved into the effect of using soft assignment distribution for different numbers of sub-networks under the self-supervised mechanism. We use DFAC-one to represent the application of soft assignment distribution to AGAT's $\widetilde{Z}$, and DFAC-all to represent the application of soft assignment distribution to all sub-networks including the embedding of AE and GAT. If set to DFAC all, the impact on performance is not significant, and it will increase computational complexity, so we set it to the configuration in DFAC.

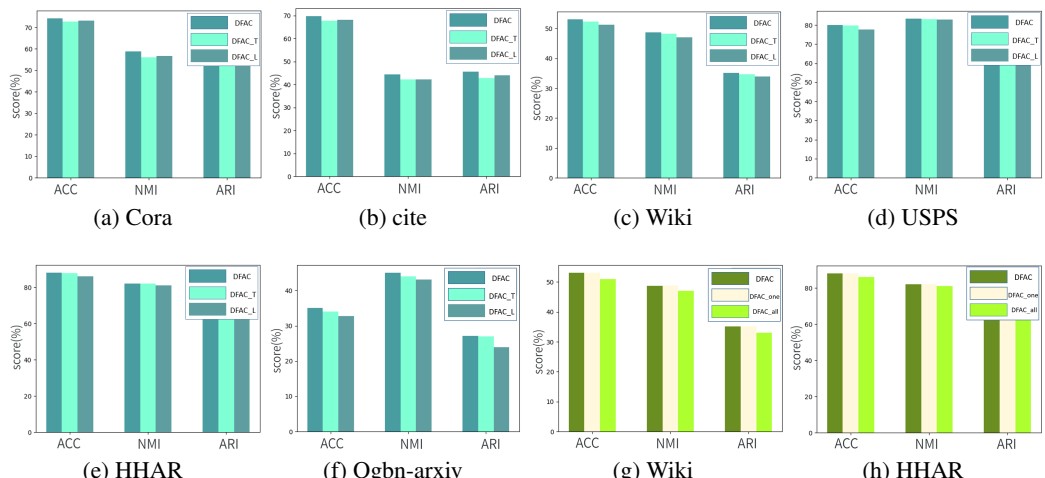

Figure 3: An in-depth exploration of the parts of the model based on the ablation experimental setup.

Table 3: Ablation study results(%)

| Methods | Cora | | | Cite | | | Wiki | | |
|---|---|---|---|---|---|---|---|---|---|
| | ACC | NMI | ARI | ACC | NMI | ARI | ACC | NMI | ARI |
| DFAC-GCN | 69.60 | 49.87 | 45.62 | 64.88 | 37.72 | 36.66 | 49.59 | 46.43 | 31.41 |
| DFAC-selfOE | 72.34 | 52.93 | 50.16 | 66.73 | 39.40 | 41.18 | 52.26 | 46.34 | 33.40 |
| DFAC-GMM | 66.40 | 51.84 | 44.23 | 64.53 | 37.89 | 38.33 | 47.10 | 44.60 | 26.84 |
| DFAC | **74.18** | **58.91** | **53.24** | **69.75** | **44.40** | **45.58** | **53.05** | **48.69** | **35.15** |

## 4.4 VISUALIZATION OF THE CLUSTERING PROCESS

In Fig 4, in order to visually verify the effectiveness of DFAC, we show the real data of the dataset, the graph-embedded data with $\alpha, \beta = 0$, and the graph-embedded data after normal clustering, respectively. DFAC can better reveal the inherent clustering structure among the data.

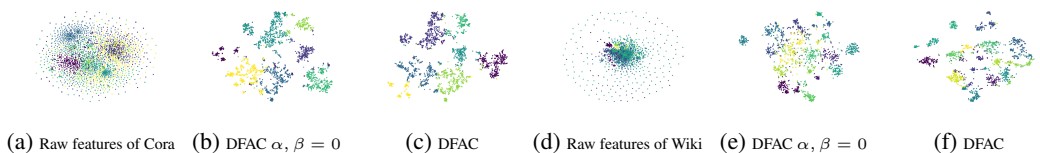

(a) Raw features of Cora    (b) DFAC $\alpha, \beta = 0$    (c) DFAC    (d) Raw features of Wiki    (e) DFAC $\alpha, \beta = 0$    (f) DFAC

Figure 4: Clustering Visualization Process and the Effect of Parameters

## 5 CONCLUSION

In this paper, we propose a new end-to-end multi-component architecture known as DFAC. We leverage the graph structure and node attributes to obtain the hidden coding of the nodes through the AGAT deep fusion network with a dual fusion mechanism in a unified framework. Based on this, using a self-optimization module and relaxed $k$-means further induces the neural network to produce graph embeddings suitable for a particular clustering model. This approach encodes more consistent and discriminative information from both sides to construct robust target distributions, which effectively improves the quality of graph embeddings. Numerous experimental results demonstrate the clustering performance of DFAC. In the future, we plan to further improve our method to adapt it to multimodal graph clustering.

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
