# OpenReview forum: "Dual Fusion AutoEncoder for Graph Clustering"
_ICLR.cc/2024/Conference — ICLR 2024 Conference Withdrawn Submission_

### Official Review · Reviewer_TvZP · 2023-10-27

**Soundness:** 2 fair
**Presentation:** 3 good
**Contribution:** 2 fair
**Rating:** 3
**Confidence:** 5

**Summary:**

The topic of this paper is deep graph clustering. The authors propose an end-to-end fusion, dual fusion autoencoder for graph clustering for deep fusion networks. It makes full use of topology and feature information of graph. Besides, it guides the network to learn relaxed k-means and perform self-supervised training. Extensive experiments demonstrate the effectiveness of the proposed method.

**Strengths:**

1. The writing is good.
2. The experiments are extensive.
3. The performance is evaluated on the middle-scale graph data, i.e., ogbn-arXiv dataset.

**Weaknesses:**

1. The motivation is not clear enough. The author should clearly point out what issue of deep graph clustering methods they want to solve.
2. The novelty is limited. The techniques have been used in various exiting methods, such as the fusion technique in DFCN [2], dual supervision training in SDCN [3], and attention mechanism in AGCN [1].
3. The survey of deep graph clustering is not comprehensive. Please refer to the survey papers [4,5]. Missing important baselines [6-11], both in the related work part and the compare experiments. The superiority is not verified since the authors miss many state-of-the-art methods.
4. The color in Figure 3 is inconsistent. In sub-figure (a)-(f), DFAC and DFAC_L use the same color. Besides, they adopt an unsightly color scheme. Sub-figures in Figure 4 are too small. In figure 1, in the bottom right box, missing the letter L.

5. Time and GPU cost experiments is missing. The authors should test the time costs and GPU costs, or provide the time and memory complexity of the proposed method.

6. To avoid the effects of randomness, the authors should run the method with different random seeds and provide the mean value and standard deviation.

[1] Peng Z, Liu H, Jia Y, et al. Attention-driven graph clustering network[C]//Proceedings of the 29th ACM international conference on multimedia. 2021: 935-943.

[2] Tu W, Zhou S, Liu X, et al. Deep fusion clustering network[C]//Proceedings of the AAAI Conference on Artificial Intelligence. 2021, 35(11): 9978-9987.

[3] Bo D, Wang X, Shi C, et al. Structural deep clustering network[C]//Proceedings of the web conference 2020. 2020: 1400-1410.
[4] Wang S, Yang J, Yao J, et al. An Overview of Advanced Deep Graph Node Clustering[J]. IEEE Transactions on Computational Social Systems, 2023.

[5] Yue L, Jun X, Sihang Z, et al. A survey of deep graph clustering: Taxonomy, challenge, and application[J]. arXiv preprint arXiv:2211.12875, 2022.

[6] Liu Y, Tu W, Zhou S, et al. Deep graph clustering via dual correlation reduction[C]//Proceedings of the AAAI Conference on Artificial Intelligence. 2022, 36(7): 7603-7611.

[7] Liu Y, Yang X, Zhou S, et al. Hard sample aware network for contrastive deep graph clustering[C]//Proceedings of the AAAI conference on artificial intelligence. 2023, 37(7): 8914-8922.

[8] Liu Y, Yang X, Zhou S, et al. Simple contrastive graph clustering[J]. IEEE Transactions on Neural Networks and Learning Systems, 2023.

[9] Liu Y, Liang K, Xia J, et al. Dink-net: Neural clustering on large graphs[J]. arXiv preprint arXiv:2305.18405, 2023.

[10] Müller E. Graph clustering with graph neural networks[J]. Journal of Machine Learning Research, 2023, 24: 1-21.

[11] Pan E, Kang Z. Beyond Homophily: Reconstructing Structure for Graph-agnostic Clustering[J]. arXiv preprint arXiv:2305.02931, 2023.

**Questions:**

See weaknesses.

---

### Official Review · Reviewer_apCG · 2023-10-30

**Soundness:** 2 fair
**Presentation:** 2 fair
**Contribution:** 2 fair
**Rating:** 3
**Confidence:** 4

**Summary:**

In this paper, the authors propose the DFAC approach to graph data, i.e., fusing feature and topological embeddings at different levels. In this paper, a self-optimizing module and a relaxed k-means loss are designed to induce more separable cluster embeddings for better performance.

**Strengths:**

The authors propose an interesting way of using AE structures to enhance the performance of GNNs. And the feasibility is verified by experiments.

**Weaknesses:**

1. Written logic needs improvement. Many sentences are difficult to understand.

2. Mathematical notation is confusing and many symbols are not specified or explained. For example, equations 4, 6.

3. The “self-supervised training module” is the almost same the famous DEC.

4. The other “proposal” is the relaxed k-means loss. But it actually is the second-order similarity measurement of the original features.

**Questions:**

1. The paper argues “a deep fusion of autoencoder (AE) and graph attention network (GAT) networks to alleviate the over-smoothing problem.” However, there is no detailed description or theory to support the argument.

2. I believe that the convergence of the total loss in DFAC is challenging to achieve. Therefore, it is not reasonable to use convergence as the termination condition for Algorithm 1.

3. The “self-supervised training module” is the almost same the famous DEC, but the authors did not provided the reference.
[1] Xie et al Unsupervised Deep Embedding for Clustering Analysis.

4. The rigour of the writing needs to be improved, for example, the paper states that "all of these methods suffer from overfitting problems". In fact, there are already many methods that are effective in alleviating the problem.

5. The ablation experiment was incomplete; for instance, the experiment concerning parameter beta was not included.

6. How are trade-off parameters like $\ell$ and $\epsilon$ learned in detail? Does this learning process result in these parameters becoming fixed at 1 or 0, leading to a loss of their usefulness?

---

### Official Review · Reviewer_kMme · 2023-11-01

**Soundness:** 3 good
**Presentation:** 3 good
**Contribution:** 2 fair
**Rating:** 3
**Confidence:** 5

**Summary:**

This paper argues that deep fusion graph clustering methods are limited by the representativeness of the chosen neural network and the choice of fusion mechanism, leading to an unpredictable degree of discretization of the learned graph embeddings. The authors propose a new end-to-end fusion, dual-fusion autoencoder for graph clustering (DFAC) for deep fusion network to obtain more compact graph embeddings compatible with clustering tasks.

**Strengths:**

1. This paper is well-written and easy to follow.

2. Performance is good. Experimental performance outperforms most comparative methods.

3. The experiment is valid. The authors have verified through experiments such as ablation experiments that the various modules of the proposed method can further improve graph representation learning and subsequent graph clustering. The effectiveness of the proposed method is proven.

**Weaknesses:**

1. Lack of innovation. The proposed approach may seem novel, but the modules are all based on existing work, i.e., DEC[1] and GAT[2]. The method just integrates them and lacks innovativeness.
[1] Unsupervised deep embedding for clustering analysis, ICML16.
[2] Graph Attention Networks, ICLR18.

2. Writing error. Although the paper is written logically and fluently, it still has many formatting and grammatical errors. For example, the last paragraph of Section 2.1 lacks a period. And The matrix representation should be bold. Table fonts should be uniform in size.

3. Lack of up-to-date comparison methods. There are only two comparison methods published after 2022, and it is recommended that more latest comparison methods be added.

4. Insufficient parameter sensitivity analyses. Although the paper performed a parameter sensitivity analysis for the main parameter $\alpha$, for parameters such as $\beta$ and $\epsilon$ only fixed assignments were made. They should have also performed parameter sensitivity analyses to further enrich the experiments and prove the validity of the method.

5. Experimental results should record the variance of the results.

**Questions:**

Can the authors elaborate on the differences with other existing work such as GAT and EGAE etc.? The method proposed by the authors just integrates them and lacks innovation and contribution.

---

### Official Review · Reviewer_UPjN · 2023-11-03

**Soundness:** 2 fair
**Presentation:** 3 good
**Contribution:** 2 fair
**Rating:** 5
**Confidence:** 5

**Summary:**

- The paper proposes a new method for clustering graphs using a deep fusion network with a dual fusion mechanism.
- The paper introduces a relaxed $k$-means and a self-supervised embedding module to optimize the graph embeddings and the clustering results.

**Strengths:**

- The paper proposes a new end-to-end fusion network for graph clustering, which can capture both graph structure and node feature information.
- The paper introduces a novel dual fusion mechanism, which can fuse cross-modal embeddings from shallow and deep layers, leading to better clustering results.
- The paper also incorporates relaxed $k$-means and self-supervised embedding modules to optimize the graph embeddings and the clustering objectives in a unified framework.

**Weaknesses:**

- Most the techniques are existing, and most of the baselines works use the same motivation. The GNN part is similar to GAT actually.
- The paper does not provide a clear explanation of how the dual fusion mechanism works, and what are the benefits of fusing cross-modal embeddings from different layers.

**Questions:**

- Highlight Uniqueness: Clearly emphasize the unique aspects or improvements of your proposed techniques compared to existing methods. What sets your approach apart from other baselines? Why is it different or better?
- Motivation Clarification: If you believe the motivation is similar to existing baselines, consider explaining how your work builds upon or extends the existing motivation. Is there a specific aspect or application where your approach excels?
- GNN Comparison: If the GNN part is similar to GAT, provide a clear comparison that outlines the similarities and differences. Explain any enhancements or modifications you've made to adapt GAT to your specific context.

**Details Of Ethics Concerns:**

NAN